# Monodisperse MoS_2_/Graphite Composite Anode Materials for Advanced Lithium Ion Batteries

**DOI:** 10.3390/molecules28062775

**Published:** 2023-03-19

**Authors:** Baosheng Liu, Feng Li, Hongda Li, Shaohui Zhang, Jinghua Liu, Xiong He, Zijun Sun, Zhiqiang Yu, Yujin Zhang, Xiaoqi Huang, Fei Guo, Guofu Wang, Xiaobo Jia

**Affiliations:** School of Electronic Engineering, Guangxi University of Science and Technology, No. 2 Wen-Chang Road, Liuzhou 545006, China

**Keywords:** lithium-ion battery anode, molybdenum disulfide, graphite, mechanical ball-milling, composite materials

## Abstract

Traditional graphite anode material typically shows a low theoretical capacity and easy lithium decomposition. Molybdenum disulfide is one of the promising anode materials for advanced lithium-ion batteries, which possess low cost, unique two-dimensional layered structure, and high theoretical capacity. However, the low reversible capacity and the cycling-capacity retention rate induced by its poor conductivity and volume expansion during cycling blocks further application. In this paper, a collaborative control strategy of monodisperse MoS_2_/graphite composites was utilized and studied in detail. MoS_2_/graphite nanocomposites with different ratios (MoS_2_:graphite = 20%:80%, 40%:60%, 60%:40%, and 80%:20%) were prepared by mechanical ball-milling and low-temperature annealing. The graphite sheets were uniformly dispersed between the MoS_2_ sheets by the ball-milling process, which effectively reduced the agglomeration of MoS_2_ and simultaneously improved the electrical conductivity of the composite. It was found that the capacity of MoS_2_/graphite composites kept increasing along with the increasing percentage of MoS_2_ and possessed the highest initial discharge capacity (832.70 mAh/g) when MoS_2_:graphite = 80%:20%. This facile strategy is easy to implement, is low-cost, and is cosmically produced, which is suitable for the development and manufacture of advance lithium-ion batteries.

## 1. Introduction

Energy is an important foundation to support the continuous development of human society. Energy scarcity, the breeding of environmental pollution, and other problems have aroused widespread concern in today’s era of high energy and high consumption, as well as rapid technological development. The large-scale use of traditional non-renewable energy sources is bound to cause environmental problems such as the ozone hole and the greenhouse effect, which seriously threaten human health [1]. In this context, lithium-ion batteries have rapidly developed into a new generation of energy-storage power supplies based on their characteristics of high specific capacity, high cycle life, small size, light weight, no memory effect, and no pollution. The traditional graphite anode material has a low theoretical capacity (theoretical capacity of 372 mAh/g [2,3,4]), and it is easy to form lithium dendrites at room temperature and low temperature, resulting in poor cyclic stability, reduced safety, and other problems that can no longer meet the demand for the high energy density and high power density of power batteries [5,6,7].

To solve the above problems, some researchers have focused on two-dimensional materials with unique structures [8,9,10], among which molybdenum disulfide is favored by researchers due to its high theoretical specific capacity (670 mAh/g), low cost, and layered structure that facilitates Li^+^ de-embedding [11]. Considering that MoS_2_ has poor electrical conductivity and is prone to agglomeration during charging and discharging, resulting in poor cycling stability, the widely adopted solution is to combine MoS_2_ with materials of strong electrical conductivity and stabilization, which has a synergistic effect and then improves the electrochemical properties, such as cycling stability. In the research work of Ma et al. [12], hierarchical carbon/MoS_2_ composites were successfully prepared via the facile hydrothermal method to enhance the electrochemical performance of MoS_2_ anodes for lithium-ion batteries. The porous structure could provide a rapid transport path for lithium-ion diffusion, which effectively improved the cycling stability of the anode material. However, the presence of porous carbon does not effectively improve the discharge capacity of the composite, which is what needs to be addressed. Hai et al. [13] prepared MoS_2_/carbon nanotube composites by a high-energy mechanical grinding method and found that the discharge-specific capacity and cycling capacity retention of the materials were significantly improved when the mass ratio of molybdenum disulfide to carbon nanotubes was 1:2, the first discharge capacity was as high as 1703 mAh/g at a current density of 100 mA/g, and the capacity retention was around 85% after 70 cycles. The introduction of MoS_2_ effectively improves the discharge capacity, and carbon nanotubes can act as a bridge between the layers of molybdenum disulfide, solving the drawback that lithium ions cannot be efficiently transported between the composite layers. The problem in this work is that multi-walled carbon nanotubes are expensive and cannot meet the requirements of low cost and high yield in the production of anode materials for lithium-ion batteries. Considering the inherent structural collapse of molybdenum disulfide during cyclic charge/discharge, many researchers have optimized the structure of molybdenum disulfide during its preparation to improve its electrochemical properties. Qi et al. [14] successfully synthesized molybdenum disulfide/reduced graphene oxide (MoS_2_/rGO) composites by the hydrothermal method. The rGO interconnections provide excellent conductive networks for the lamellar structure and can act as a carrier for the growth of molybdenum disulfide nanosheets to achieve structural support. But this work may produce atomic-scale defects in the substrate when preparing graphene oxide dispersion by the Hummer method, which will have an impact on the morphology of the composites [15].

Among many material compounding processes, carbon cladding modification is a very common means of surface modification that can significantly improve the electrical conductivity of the material. Jin et al. [16] prepared a carbon-coated molybdenum disulfide (C@MoS_2_) composite material for lithium-ion battery anode material by the hydrothermal in situ method. Studies have shown that the composite material has a porous carbon-coated structure and good electrochemical performance. After 200 cycles at the current density of 500 mA/g, the capacity retention rate is as high as 94%. But the current carbon cladding process and the preparation process of advanced carbon materials are relatively complex, and the production cost of electrode materials prepared on this basis is high, which is still not conducive to commercial application. In addition to the above methods, nanosizing MoS_2_ materials is also an effective way to improve their cycling performance, but nanoparticles are characterized by both low first-cycle Coulomb efficiencies and complicated preparation processes. Therefore, the research on the technology of micro and nano composite structures (microscale particles with built-in nanoscale features) of materials will be very promising after satisfying the conditions of commercial feasibility such as safety and cost [17].

In comparison to the previous research, compounding the most widely used graphite anode material in the industry with two-dimensional molybdenum disulfide materials, while providing sufficient electrical conductivity and depolarization, is conducive to a reduced technology route in the battery industry to achieve graphite anode replacement at a lower cost [18]. Based on this idea, in this paper, MoS_2_/graphite composites were prepared from monodisperse nano-layered molybdenum disulfide [19] and conventional graphite anode materials by mechanical ball-milling, and the ball-milling process resulted in a uniform distribution of graphite sheets between MoS_2_ sheets, which both slowed down the agglomeration of MoS_2_ and improved the electrical conductivity of the composites, showing excellent electrochemical properties overall. This method provides new ideas for the design and optimal modification of high-performance lithium-ion battery cathode materials, which can effectively contribute to the rapid development of new energy vehicle power battery technology research.

## 2. Results and Discussion

### 2.1. Structure and Morphology of MoS_2_/Graphite Composites with Different Ratios

The X-ray diffractograms of the anode materials with different composite ratios of MoS_2_/graphite are shown in Figure 1a, and it can be seen that all four composite anode materials show a crystal structure similar to the intrinsic MoS_2;_ the diffraction peaks shown in the figure (14.4° (002), 32.7° (100), 39.3° (103), 49.8° (105), and 58.3° (110)) all correspond to the diffraction peaks of the hexagonal crystal structure of molybdenum disulfide [20]; and the peaks appear in perfect agreement when compared with the standard card PDF # 37-1492. Moreover, the peak positions and peak shapes of all diffraction peaks are clear, indicating that the composite anode material has high crystallinity and purity. In addition, a diffraction peak of varying intensity was observed at 26.5° in the diffractograms of the four MoS_2_/graphite composites with different ratios, which proved the presence of graphite and coincided with the expected results that the intensity of this peak was decreasing as the percentage of graphite material in the composites decreased. In contrast, the intensity of the MoS_2_ characteristic peak does not increase as its percentage increases, because the change in the intensity of the MoS_2_ characteristic peak reflects the change in the number of layers of the two-dimensional material, which indicates that the number of layers of the MoS_2_ material does not change after ball-milling [21]. No other obvious spurious peaks were detected, indicating that no impurities were generated and the original structure of the composite was not destroyed during the ball-milling process. In addition, it can be seen in Figure 1b that the characteristic peak of the MoS_2_ material at 14.6° (2θ) has significantly shifted to the left after ball-milling, which further indicates that the ball-milling process may have made the crystallite size of the MoS_2_ material larger. In contrast, the characteristic peak at 26.6° (2θ) of the graphite material after ball-milling has slightly shifted to the right, which indicates that the ball-milling process has no significant effect on the crystallite size of the graphite material.

The SEM images of graphite, intrinsic MoS_2_ and MoS_2_/graphite composite anode materials are shown in Figure 2. The morphology of the intrinsic graphite material is a flattened sphere with a particle size of 9 μm (Figure 2a), and the morphology of the intrinsic molybdenum disulfide material is a monodisperse nanolayer (Figure 2b), which has the advantages of uniform particle size, high activity, a large specific-surface area, and a high sample-adsorption capacity [22,23]. By comparison, the morphology of the MoS_2_/graphite composites was found to be significantly different from that of the intrinsic graphite material (Figure 2a) and the intrinsic molybdenum disulfide material (Figure 2b) [24]. The graphite in its original flattened spherical form is dispersed into thin flake layers by a ball-milling process [25], and the molybdenum disulfide is simultaneously ball-milled into flake structures. The lamellar graphite effectively enters between the MoS_2_ flakes and mixes uniformly to form compacted aggregates. By comparing Figure 2c–f, it can be found that with the increasing percentage of MoS_2_, it can be observed that the lumpy particles are reduced and MoS_2_ is uniformly dispersed. When the particle size of the graphite material is significantly reduced and the percentage of molybdenum disulfide reaches 80%, the particle size of the composite is mostly between 1–2 μm. By observation, Figure 2f shows mostly flaky MoS_2_ material, whose discharge capacity is expected to be enhanced by a large amount of molybdenum disulfide. Overall, the compounded MoS_2_/graphite aggregates can effectively increase the contact area between the material and electrolyte, alleviate the agglomeration phenomenon, and improve the discharge capacity, but the excessive MoS_2_ will reduce the cycling stability.

To further characterize the dispersion of MoS_2_ material in MoS_2_/graphite composites, EDS tests were conducted on four MoS_2_/graphite composites with different ratios, and the test results are shown in Figure 3. It is easy to see that with the increasing percentage of MoS_2_, the Mo elements and S elements in the composites become more and more obvious, and in addition, it is obvious through the four graphs in Figure 3 that the elements such as Mo, S, and C in the composites are uniformly distributed without any abnormal phenomena such as agglomeration, which indicates that the MoS_2_ materials are very uniformly compounded with graphite materials and play a positive role in improving the electrochemical performance of the anode materials [26].

### 2.2. Electrochemical Properties of MoS_2_/Graphite Composites with Different Proportions

Initial charge–discharge curves, various current-capacity tests, and cycling tests (0.05 C/1 C) were conducted on the prepared coin-cell in the voltage range of 0.05–3.00 V to study the electrochemical performance of MoS_2_/graphite anode materials with different composite ratios. The test results with the contrast of graphite anode are shown in Figure 4.

Figure 4a shows the activation performance of the MoS_2_/graphite composite anode materials under the same test conditions, and the first discharge capacities of the MoS_2_/graphite (20%:80%, 40%:60%, 60%:40%, and 80%:20%) composites were 542.127 mAh/g, 615.768 mAh/g, 787.463 mAh/g, and 832.699 mAh/g, respectively. The first discharge capacity of MoS_2_/graphite composite increases as the percentage of molybdenum disulfide increases, with the highest discharge capacity, achieved when the percentage of molybdenum disulfide was 80%, which is approximately twice that of the intrinsic graphite material. Therefore, it can be concluded that the increase in the capacity of MoS_2_/graphite composites is attributed to the presence of elemental sulfur in molybdenum disulfide [27]. Graphite plays an important role in making up the difference in the conductivity of molybdenum disulfide in composites. The first Coulomb efficiencies of the MoS_2_/graphite (20%:80%, 40%:60%, 60%:40%, and 80%:20%) composites with different ratios were 97.50%, 93.84%, 86.23%, and 87.84%, showing a decreasing trend due to the higher percentage of molybdenum disulfide in the composite; the more obvious is its volume expansion and structural collapse [28].

Cycle performance is an important indicator to measure the electrochemical performance of lithium-ion batteries [29]. Figure 4b shows that MoS_2_:graphite = 20%:80% and MoS_2_:graphite = 40%:60%, MoS_2_:graphite = 60%:40%, MoS_2_:graphite = 80%:20% in the voltage range of 0.05–3.00 V, the performance curve of 300 cycles of constant current charge/discharge under the condition of 1 C high rate. The results show that, with the increase in the proportion of MoS_2_, the cycle performance of the composite material shows a trend of first increasing and then decreasing. After 300 cycles, when MoS_2_:graphite = 40%:60%, the cycle-capacity retention rate is the best, and the cycle capacity is the highest at 58.328 mAh/g with 99.89% coulombic efficiency, showing that this composite ratio of electrodes has excellent reversibility. When MoS_2_:graphite = 80%:20%, the initial capacity of the composite material is the lowest at 19.936 mAh/g, and the cycle capacity retention rate is also the worst (only 21.12%), which is far lower than the discharge capacity of the intrinsic graphite anode material. The reason is that when charging and discharging under high-rate conditions, due to the excessive proportion of molybdenum disulfide in the MoS_2_/graphite composite material, the structure of the composite material is destroyed along with the intercalation and deintercalation of Li^+^ [30].

Figure 4c depicts the performance curves of the four composite materials under constant current charge and discharge conditions at 0.05 C. It can be concluded that the cycle capacity retention rate of MoS_2_/graphite (20%:80%, 40%:60%, 60%:40%, and 80%:20%) composites shows a downward trend with the increase of the proportion of MoS_2_ (99.80%, 85.67%, 79.97%, and 36.83%), and when MoS_2_:graphite = 80%:20%, the capacity of the composite material is already much lower than that of the intrinsic graphite anode material. When MoS_2_:graphite = 20%:80%, the discharge-specific capacity does not seem to change after cycling, and the cycle-capacity retention rate is as high as 99.80%, with 99.93% coulombic efficiency, which shows that a large number of graphite sheets are evenly distributed between the molybdenum disulfide sheets, which stabilizes the MoS_2_/graphite composite structure and enabling the electrode to obtain excellent reversibility. When MoS_2_:graphite = 80%:20%, the cycle performance is the worst, and the capacity retention rate is only 36.83%. Along with the decreasing percentage of C elements in MoS_2_/graphite materials, the electrical conductivity and structural stability of the composites will continue to deteriorate, which is the reason for the decreasing trend of the cyclic capacity retention of the composites [31].

For the practical application of lithium batteries, electrodes must have excellent reversible rate performance. The rate capability of both graphite and MoS_2_/graphite electrodes was measured, ranging from low to high charge/discharge rates, i.e., 0.05, 0.1, 0.2, 0.5, and 1 C (Rated current capacity, and 1C charging or discharging means that the battery is fully charged or fully discharged within 1 h.), as illustrated in Figure 4d. It can be found that when the charge and discharge test is performed under low-rate conditions, the composite material with a small proportion of molybdenum disulfide has a higher discharge capacity. When the charge/discharge rate increases, the MoS_2_/graphite composite material shows poor stability, and its cycling capacity also shows a downward trend, which shows that the MoS_2_/graphite composite material has poor adaptability to the change of charge/discharge current. The reason for this problem is inseparable from the characteristics of the molybdenum disulfide material itself, such as poor conductivity and an unstable structure during cycling. Solving this problem will be our next research project.

In order to explore the reasons for the decrease in the specific capacity of the graphite electrode and MoS_2_/graphite composite electrode, SEM tests were conducted on the pole pieces before and after 300 cycles at 1 C rate, and the test results are shown in Figure 5. We further analyzed that the graphite electrode flakes exhibit the morphological characteristics that the conductive agent uniformly adheres to the graphite material surface before cycling and the contact between them is excellent (Figure 5a); the MoS_2_/graphite composite electrode flakes exhibit a clear two-dimensional lamellar monodisperse structure, and this structure also allows the close contact between the active material and the conductive agent (Figure 5b) [32]. The decrease in the specific capacity of graphite electrode sheet after cycling can be attributed to the formation of a thick and rough SEI layer covering the surface of electrode material by the side reaction with electrolyte (Figure 5c) [33,34]; meanwhile, along with the cyclic charging and discharging, more obvious cracks appear on the surface of graphite electrode (Figure 5e), which will be very unfavorable to the contact between the active material and conductive agent [35]. For the MoS_2_/graphite composite electrode, the irreversibly grown SEI layer that forms is thinner and more stable (Figure 5d), but the formation process also consumes a large amount of electrolyte, which leads to the electrolyte drying phenomenon and further decreases the specific capacity of the MoS_2_/graphite composite electrode [36,37]. In addition, some cracks were also observed on the surface of the MoS_2_/graphite composite electrode after cycling (Figure 5f), and these cracks would deteriorate the contact between the active substance and the conducting agent, thus causing a sharp decrease in the specific capacity of the MoS_2_/graphite composite electrode.

Figure 6a shows the cyclic voltammetry test curves of intrinsic graphite materials and MoS_2_/graphite composites with four different ratios in the voltage range of 0.05–3.00 V. For the first cathodic process, the MoS_2_/graphite electrode depicts cathodic peaks between 0.75–1.25 V and 1.50–2.00 V. The reduction peaks between 1.50 and 2.00 V can be attributed to the intercalation of Li^+^ into the layered structure of MoS_2_ to form Li_x_MoS_2_. The peak between 0.75 and 1.25 V is attributed to the conversion of Li_x_MoS_2_ [38] into metallic Mo and Li_2_S, respectively. The reaction process is shown in reaction Equation (1) and reaction Equation (2) [39].
MoS_2_+xLi^+^ + xe^−^→Li_x_MoS_2_(1)
Li_x_MoS_2_ + (4 − x)Li^+^ + (4 − x)e^−^→Mo + 2Li_2_S(2)
 Li_2_S→2Li^+^ + S + 2e^−^(3)

In the reverse anodic process, the MoS_2_/graphite composite material has an obvious oxidation peak in the voltage range of 2.25–3.00 V, which may be related to the oxidation of Li_2_S to S; the reaction process is shown in reaction Equation (3). In addition, a series of redox peak pairs between 0.05 and 0.27 V is related to the intercalation and exportation of Li^+^ on graphite interlayer carbon atoms.

The electrochemical impedance spectra (EIS) of graphite and MoS_2_/graphite are shown in Figure 6b. The intercept of the curve at high frequency with the axis corresponds to electrolyte resistance and contact resistance. The inset of Figure 6b is the equivalent circuit of an EIS fitting; R_1_ is the ohmic resistance of the electrode, and R_2_ is the charge transfer resistance. CPE is an abbreviation for the constant phase elements. W_1_ is the Warburg impedance [40]. The R_1_ value of the graphite and MoS_2_:graphite = 20%:80%, 40%:60%, 60%:40%, and 80%:20% electrodes is calculated to be 4.73 Ω, 4.09 Ω, 5.39 Ω, 3.84 Ω, and 4.16 Ω, respectively, indicating that the ohmic resistance of the above electrodes in this study were very close and the errors in the preparation process were relatively small. According to the inserted equivalent circuit [41], the R_2_ value of the graphite and MoS_2_:graphite = 20%:80%, 40%:60%, 60%:40%, and 80%:20% electrodes is calculated to be 19.57 Ω, 16.04 Ω, 106.89 Ω, 74.97 Ω, and 428.64 Ω. When MoS_2_:graphite = 20%:80%, the charge-transfer impedance is the smallest, and its slope in the low-frequency region is the largest, indicating that the diffusion resistance of Li^+^ in the electrode at this time is the smallest. It can be concluded that an appropriate amount of MoS_2_ combined with a graphite material with better conductivity can have a synergistic effect to reduce the charge transfer resistance of the composite material and improve the diffusion efficiency of Li^+^ [42].

## 3. Experiment

### 3.1. Preparation of Anode Materials

In this experiment, 0.45 g of sodium molybdate dihydrate (Na_2_MoO_4_·2H_2_O, Aladdin Reagent Co., Shanghai, China) and 0.6 g of thiourea (CH_4_N_2_S, Aladdin Reagent Co., Shanghai, China) were first mixed in 36 mL of deionized water (Molecular Lab water ultra-purifier, Shanghai, China) as precursors, and the well-mixed solution was transferred into a 100 mL Teflon-lined stainless steel autoclave (Hefei kejing materials technology co., LTD, Hefei, China) and sealed tightly, and heated at 220 °C for 24 h in a furnace [43,44]. After cooling naturally, the black precipitates were collected by centrifugation, washed with deionized water and anhydrous ethanol (C_2_H_5_OH, Sinopharm Chemical Reagent Co., Shanghai, China), and dried in a vacuum oven at 80 °C for 24 h [45,46]. The as-prepared MoS_2_ was annealed in a conventional tube furnace at 500 °C for 2 h in a stream of argon gas (Ar, Liuzhou Xineng Gas Co., Liuzhou, China) [47]. Finally, monodisperse nano-layered molybdenum disulfide was obtained. The graphite anode material was selected from the 918 series of natural graphite (C, Tianjin Battery Co., Tianjin, China) with excellent cycling performance and multiplicative performance [48,49]. The prepared molybdenum disulfide powder and natural graphite were mixed in four different ratios of MoS_2_:graphite = 20%:80%, 40%:60%, 60%:40%, and 80%:20% to investigate the effect of different MoS_2_ contents on the anode performance. The four ratios of anode materials were put into four agate ball-milling jars, and anhydrous ethanol was poured in to avoid oxidation of the anode materials during the ball-milling process until the anhydrous ethanol covered the agate balls in the ball-milling jars [50,51]. The four different proportions of composite materials were ball-milled at low speed at 50 Hz for 2 h to fully mix them, then placed in a blast drying oven at 60 °C for 24 h. After all the anhydrous ethanol evaporated, the composite powder was collected.

### 3.2. Preparation of Button Cell

The 0.8g of MoS_2_/graphite composite anode material, 0.1 g of super P (Sinopharm Chemical Reagent Co., Shanghai, China), and 0.1 g of CMC (Carboxymethyl Cellulose Sodium, Shanghai Macklin Biochemical Co., Shanghai, China) were poured into the weighing bottle according to the mass ratio of 8:1:1, and the appropriate amount of deionized water and N-Methylpyrrolidone (NMP, Shanghai Aladdin Biochemical Co., Shanghai, China) was added according to the fluidity state of the material for stirring [52], and then it was stirred into a slurry with good fluidity and evenly coated on the matte surface of the clean copper foil (Hefei kejing materials technology Co., Ltd.), and then it was put into a vacuum oven and dried at 120 °C for 8 h [53]. After rolling, copper foil coated with anode material was cut into 12 mm-diameter round anode sheets by a cutting machine, and then the round anode sheet was weighed (the mass of the active material used in the electrode sheet was about 11 mg). After weighing, the anode sheet was placed in a paper package and placed in a vacuum-drying oven for 8 h at 120 °C. The average areal mass loading of the active materials was around 2.43 mg/cm^2^. To ensure the influence of the environment on the preparation process of anode material, the anode material preparation process should be carried out in an environment with a humidity of 20% or less.

The standard button cells (CR2025) were assembled in an argon-filled glovebox (O_2_ ≤ 0.01 ppm and H_2_O ≤ 0.01 ppm), using MoS_2_/graphite electrodes as working electrodes and lithium metal chips (0.45 × 15.6 mm, Neware Technology Ltd., Shenzhen, China) as the counter electrode. Two insulating separators (2400 Celgard PP membrane) were used to prevent short-circuiting of the cell. The electrolyte (Jiangxi Jinhui Lithium Electric Materials Co., Ltd., Jiangxi, China) was composed of 1 M LiPF_6_ in ethylene carbonate (EC), diethyl carbonate (DEC), and ethyl methyl carbonate (EMC) mixed solution (EC: DEC: EMC = 1:1:1 vol%) [54].

### 3.3. Material Structure and Morphology Characterization Testing

The crystal atomic structure of the sample to be tested was determined by an XRD (X-ray diffraction) test, and the phase composition of the material was analyzed qualitatively and quantitatively. The test instrument was Rigaku SmartLab SE, Japan. The test range was 5–90°, the scan rate was 5°/min, and the light source was Cu-Kα ray. The tube voltage was 40 kV, the tube current was 40 mA, and the scanning mode was continuous scanning. The TESCAN MIRA LMS (Czech Republic) was used for scanning electron microscopy (SEM) testing to observe the surface morphology, smoothness, and particle size of the cathode material at the microscopic level. Qualitative and quantitative analysis of elements can be performed by analyzing the types and compositions of elements with energy-dispersive X-ray spectroscopy (EDS).

### 3.4. Electrochemical Performance Testing

The previously assembled button cell in the glove box was transferred into the thermostat (thermostat temperature was 25 °C), and the constant current charge/discharge performance test was conducted in the test voltage range of 0.05–3.00 V (the test instrument was selected from Shenzhen Xinwei Battery Testing System BTS4000). AC impedance test (EIS) and cyclic voltammeter test (CV) needed to be tested on the electrochemical workstation (model CHI760E), with a test voltage range of 0.05–3.00 V, AC impedance spectrum test set frequency range of 0.01–100 000 Hz, the amplitude of 5 mV, bias current below 0.01 Hz, using Fourier transform. The scanning speed of the cyclic voltammeter test was 0.1 mV/s.

## 4. Conclusions

In this study, the outstanding electrochemical performance of MoS_2_/graphite composites could be attributed to the following reasons: After ball-milling, the MoS_2_ sheets and graphite sheets are uniformly dispersed. The presence of molybdenum disulfide enhances the overall capacity, and the graphite sheets effectively alleviate the agglomeration of MoS_2_ and improve electrical conductivity. As evidenced by characterization testing, the MoS_2_/graphite composite anode material has high crystallinity and purity. The electrochemical performance test shows that the capacity of MoS_2_/graphite composites possessing the highest first discharge capacity occurred when MoS_2_:graphite = 80%:20%; after 300 cycles of charge/discharge at high magnification (1 C), it was found that the highest reversible capacity occurred when MoS_2_:graphite = 40%:60%; the cyclic charge/discharge test results at a low rate (0.05 C) showed that the MoS_2_/graphite composites showed the best cycling performance when MoS_2_:graphite = 20%:80%. To sum up, different composite proportions can improve the different electrochemical performances of the battery. The performance research and analysis of four different proportions of MoS_2_/graphite composite materials in this paper demonstrate a promoting and guiding role in the further optimization and development of high-performance lithium-ion battery anode materials.

## Figures and Tables

**Figure 1 molecules-28-02775-f001:**
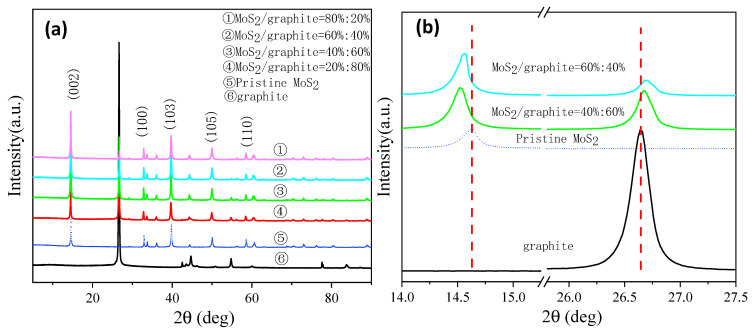
(**a**) XRD patterns of MoS_2_/graphite composites at different proportions; (**b**) Partial XRD patterns.

**Figure 2 molecules-28-02775-f002:**
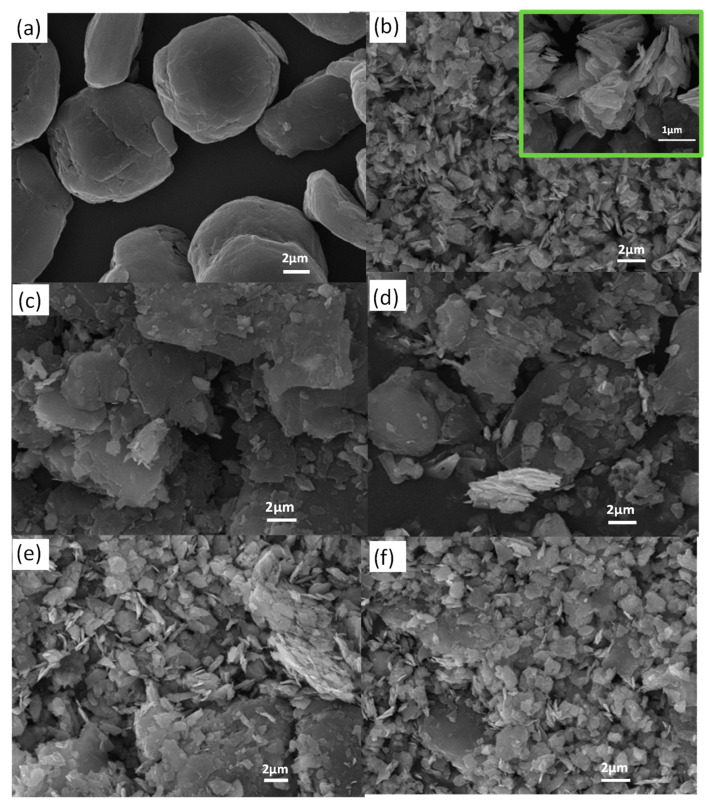
SEM images: (**a**) Graphite; (**b**) MoS_2_ (The image obtained after zooming in is shown in the green box); (**c**) MoS_2_/graphite (20%:80%); (**d**) MoS_2_/graphite (40%:60%); (**e**) MoS_2_/graphite (60%:40%); (**f**) MoS_2_/graphite (80%:20%).

**Figure 3 molecules-28-02775-f003:**
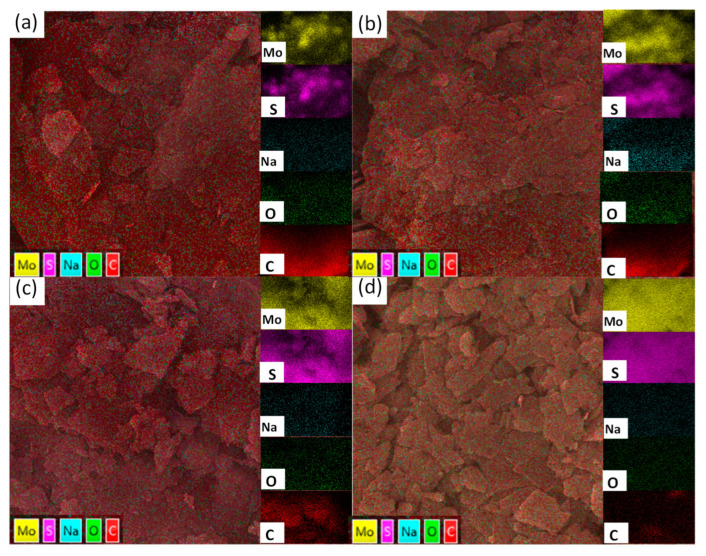
EDS images: (**a**) MoS_2_/graphite (20%:80%); (**b**) MoS_2_/graphite (40%:60%); (**c**) MoS_2_/graphite (60%:40%); (**d**) MoS_2_/graphite (80%:20%).

**Figure 4 molecules-28-02775-f004:**
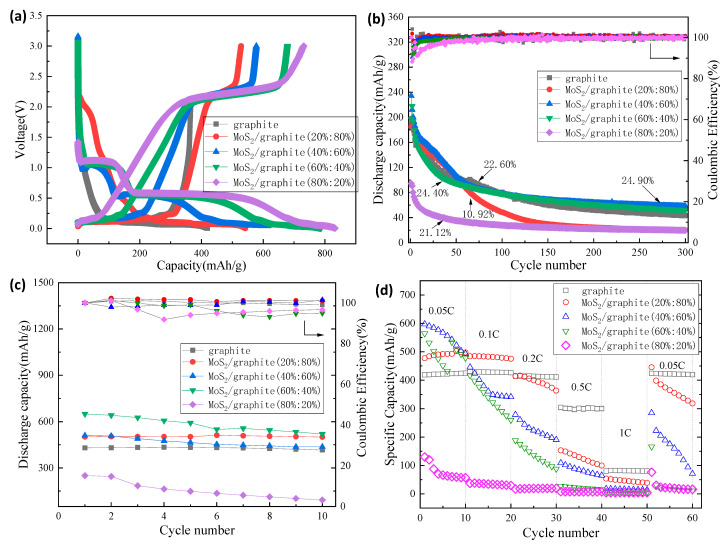
(**a**) Initial charge–discharge capacity curve; (**b**) 1 C cycling; (**c**) 0.05 C cycling; (**d**) Rate capability curve.

**Figure 5 molecules-28-02775-f005:**
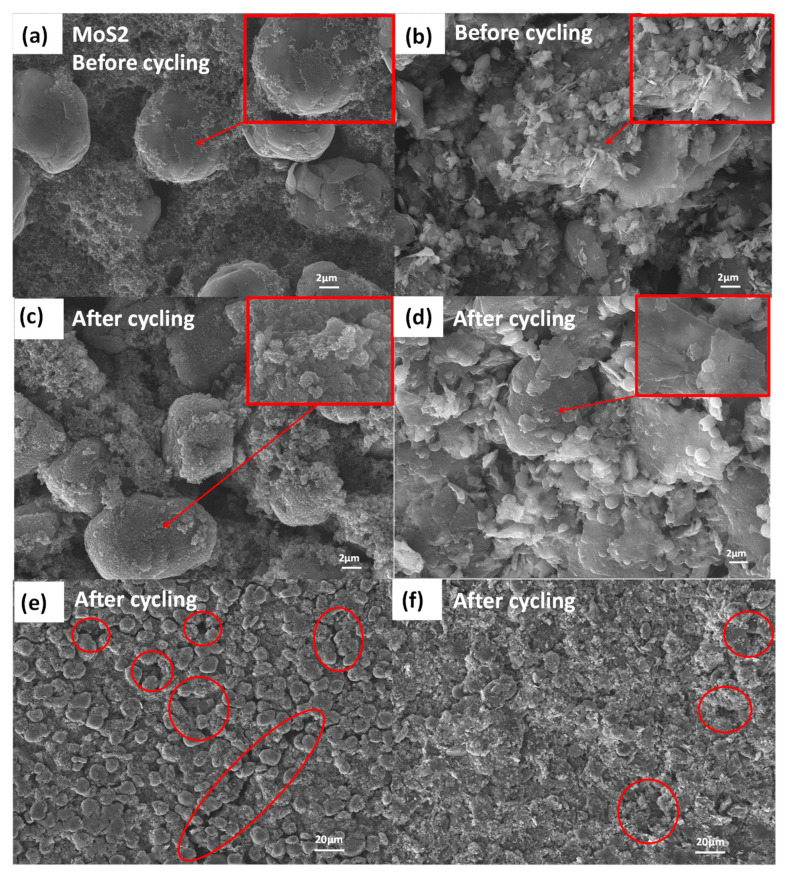
SEM image of the electrode before and after 300 cycles at 1C rate: (**a**) Graphite electrode before cycling; (**b**) MoS_2_/graphite electrode before cycling; (**c**,**e**) Graphite electrode after cycling; (**d**,**f**) MoS_2_/graphite electrode after cycling.

**Figure 6 molecules-28-02775-f006:**
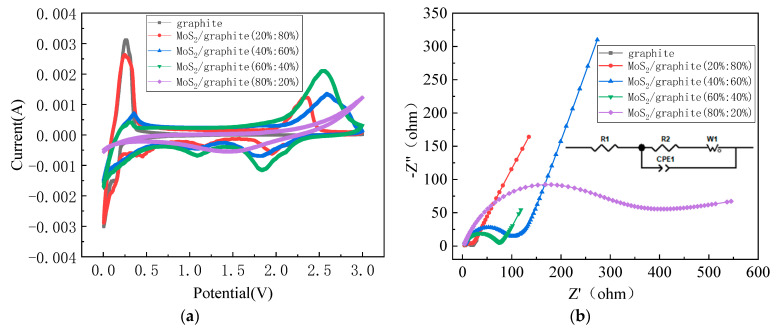
(**a**) CV test curves; (**b**) EIS test curves.

## Data Availability

The data presented in this study are available on request from the corresponding author.

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
