# Peer review of "Monodisperse MoS2/Graphite Composite Anode Materials for Advanced Lithium Ion Batteries"

_molecules, 2023, doi:10.3390/molecules28062775_

Round 1

Reviewer 1 Report

The authors had presented their work on ball milled MoS2/graphite composite as anode materials for electrochemical lithium ion battery. However, some of the corrections must be made before accepted.

1. For the experimental part, the author should provide the list of specifications of materials used in this study.

2. The authors had provided the information of the ratio of active materials used in this study. But what is the mass of active material used in the assembled button cell?

3. What is the cathode material and electrolyte used in the assembled button cell?

4. For the XRD results, is there any change in the crystallite size of graphite and MoS2 after the ball milling?

5. Figure 4b, the negative discharge capacity y-scale show in the graph should be remove. The arrow label should show the capacity retention in percentage rather than repeating the legend label. 

6. The characterisation of the MoS2/graphite composite after the electrochemical cycling should be added to explain the drastic drop of specific capacity.

7. "According to the inserted equivalent circuit [35], the R2 value of the graphite and MoS2/graphite electrodes is calculated to be 19.57 Ω, 16.04 Ω, 106.89 Ω, 74.97 Ω, 428.64 Ω." This sentence is confusing because it is not labelled according to the sample name.

8. The conclusion is too lengthy.

Reviewer 2 Report

This is a nice study on the electrochemical performance of monodisperse MoS2/graphite composite anode materials in lithium-ion batteries. This area is hot- and many people are doing similar research.  The milling technique is interesting. The study contributes to the growing research on advanced electrode materials for lithium-ion batteries. The authors presented their findings in a clear and concise manner, and the information provided appears to be consistent with their research objectives.

A few comments: 

1. Any reason the authors decide to use ‘negative electrode materials’  instead of ‘anode material’ ? 

2. The figures are blurry, and it’s difficult to read the text from the figure, including the scale bar in Fig  2, the legend in Fig 3 and 4.  Please remake the figure with a higher resolution.  

3. Did the author calculate the coulombic efficiency? That would provide meaningful info about how reversible the electrode is. 

Reviewer 3 Report

This paper can be accepted after addressing following critical issues:

(1) The language needs to be thoroughly polished, it seems to me that lots of expressions are directly translated from Chinese to English, for example, circulation capacity retention.

(2) What is the mass loading of electrodes?

(3) MoS2 is charged to 3.0V, if only charged to 2.0V/1.5V, the capacity is less than 400, is it still promising as an anode material? Commercial graphite has a theoretical capacity of 372 mAh/g and can be stably cycled for thousands of times. Comparatively, what are your advantages of MoS2/ graphite composites?

(4) Nanocomposites are challenging to be used in practical batteries, as discussed in recent review papers, Nat Rev Mater 7, 736–746 (2022), please comment on this.

(5) What is the low-temperature performance of MoS2/graphite composites? Adv. Mater. 202234, 2107899.

Reviewer 4 Report

Monodisperse MoS2/graphite composite negative electrode materials for advanced lithium ion batteries

Major revision:

In this draft, authors declared the Monodisperse MoS2/graphite composite negative electrode materials for advanced lithium ion batteries.

Comments:

1.     Authors must add the reference in the synthesis part.

2.     The synthesis part is not clear. Authors should add all the steps carefully for the readers

“In this experiment, sodium molybdate dihydrate and thiourea were first mixed in deionized water as precursors, and the well-mixed solution was transferred into the liner of the reactor and hydrothermally reacted at 220 °C for 24 h. Then the product of the hydrothermal reaction was annealed at 500 °C for 2 h under argon, and finally, monodisperse nano-layered molybdenum disulfide was obtained.”

How to dry after the first step

3.     All the figures are not clear and unable to understand both the drawing stuff and writing stuff.

Should be clear so the reviewer can understand the new findings.

4.     Conclusion is too long and not up to the mark.

Round 2

Reviewer 1 Report

The authors had made the changes based on the comments given.

Reviewer 3 Report

It can be accepted now.